# Identifying Hypertrophic or Dilated Cardiomyopathy: Development and Validation of a Fine-Tuned ResNet50 Model Based on Electrocardiogram Image

**DOI:** 10.3390/bioengineering12030250

**Published:** 2025-02-28

**Authors:** Jiayu Xu, Bo Chen, Weiyang Liu, Wei Dong, Yan Zhuang, Peifang Zhang, Kunlun He

**Affiliations:** 1Graduate School, Chinese People’s Liberation Army General Hospital, Beijing 100853, China; 2Medical Innovation Research Division, Chinese People’s Liberation Army General Hospital, Beijing 100853, China; 3Cardiology Department, Sixth Medical Center of Chinese People’s Liberation Army General Hospital, Beijing 100853, China; 4Biomind Technology Inc., Beijing 101300, China

**Keywords:** artificial intelligence, electrocardiogram image, hypertrophic cardiomyopathy, dilated cardiomyopathy, diagnosis

## Abstract

There is no established detecting tool for hypertrophic cardiomyopathy (HCM) and dilated cardiomyopathy (DCM). This study aimed to develop a deep-learning-based model for identifying HCM and DCM using standard 12-lead electrocardiogram (ECG) images. We obtained a cohort of patients with HCM (171 ECG images) or DCM (364 ECG images), confirmed by cardiovascular magnetic resonance (CMR) examinations, who underwent both ECG and CMR within 30 days at our institution. Age- and sex-matched healthy controls (2314 ECG images) were selected from our Health Check Center. A total of 2849 ECG images were processed via a fine-tuned ResNet50 architecture, with stratified five-fold cross-validation for model training, validation, and testing. The proposed model demonstrated strong performance in distinguishing DCM, achieving an area under the receiver operating curve (AUROC) of 0.996 and an area under the precision–recall curve (AUPRC) of 0.940. For the detection of HCM, the model also achieved an AUROC of 0.980 and an AUPRC of 0.953, respectively. The model prospectively exhibited stability in temporal validation. Furthermore, representative images of the Gradient-weighted Class Activation Mapping (Grad-CAM) technique analysis showed the regions corresponding to the anterior and anteroseptal leads were the most important areas for the prediction of HCM or DCM. This temporally validated fine-tuned ResNet50 model shows promise to inexpensively detect individuals with HCM or DCM.

## 1. Introduction

Hypertrophic cardiomyopathy (HCM) and dilated cardiomyopathy (DCM) are two primary types of cardiomyopathies, leading to a substantial and increasing morbidity/mortality burden. While echocardiography, genetic testing, and cardiovascular magnetic resonance (CMR) have revolutionized diagnostic precision (CMR specificity > 90% for HCM), their limited accessibility creates critical care disparities: low-income countries lack advanced cardiac imaging infrastructure [1], and rural healthcare facilities cannot perform genetic subtyping.

Electrocardiogram (ECG) remains the frontline diagnostic modality for cardiomyopathies. Not only does it have a low price and a wide range of applications, but it also provides details related to morphology, function, and genetic substrates in cardiomyopathies. Most studies on ECG in DCM are single-center retrospective studies. For example, DCM can present various ECG abnormalities [2]. However, ECG does not have sufficient diagnostic accuracy to supplant the central role of CMR for diagnosis [3]. More than 90% of HCM patients have ECG abnormalities at their first visit. ECG feature abnormalities may be the earliest indication of HCM, forming the basis for its diagnosis and treatment [1,4]. Additionally, ECG can predict adverse arrhythmic events in HCM [5,6,7,8]. However, there is no established convenient detecting tool for hypertrophic cardiomyopathy (HCM) and dilated cardiomyopathy (DCM).

Artificial intelligence (AI) has the potential to improve and optimize the clinical practice of cardiomyopathies remarkably. Algorithms modeled using raw ECG signals have been shown to detect cardiac disease [9]. Recent advances in deep learning have enhanced the etiological differentiation of cardiomyopathies. Haimovich et al. developed a Convolutional Neural Network (CNN) based on 12-lead Electrocardiogram (ECG) data, achieving an Area Under the Curve (AUC) of 0.92 for distinguishing hypertrophic cardiomyopathy (HCM) from other causes of left ventricular hypertrophy [10]. Tayal et al. implemented a random forest model integrating clinical, genetic, and CMR to subclassify dilated cardiomyopathy (DCM) into fibrosis-metabolic and non-fibrotic subtypes [11]. Shrivastava et al. demonstrated a CNN model with 98.8% sensitivity and a 100% negative predictive value for DCM detection (LVEF ≤ 45%), which was validated across multiple cohorts (AUC 0.955–0.98) [12]. Zhou et al. employed XGBoost with clinical/echocardiographic data to differentiate ischemic vs. nonischemic DCM (AUC 0.934) [13]. Comparative machine learning analyses demonstrated that a random forest model, optimized for minimum redundancy and maximum relevance, achieved a 91.2% accuracy in distinguishing between DCM, HCM, and healthy subjects, with an average AUC of 0.98.

However, the raw signal ECG data are difficult to save and do not conform to actual clinical care. To meet the observational requirements of clinicians, some research endeavors have been undertaken to analyze visual representations derived from ECG signals for the detection of heart diseases [14]. Huang et al. employed a 2D-CNN for five-class arrhythmia classification, achieving an accuracy of 99.00%, which demonstrated significant advantages over 1D-CNN models using raw ECG signals (90.93% accuracy) [15]. This approach eliminates the need for manual preprocessing (e.g., signal filtering and feature selection), validating the superiority of two-dimensional representations. Diker et al. and Ullah et al. further applied a short-time Fourier transform (STFT) to an ECG time–frequency analysis: the former study conducted comparative analyses of VGG-16, ResNet-18, and AlexNet architectures, demonstrating superior performance, with AlexNet achieving optimal results [16,17].

Despite the rapid increase in the understanding of the genotype–phenotype correlations in DCM/HCM and the role of the ECG, the knowledge of ECG images of the disease is still limited. In this study, we develop a ResNet50-driven framework for differentiating HCM or DCM from standard 12-lead 10 s ECG images, addressing gaps in detecting tools for cardiomyopathies. We developed, tested, and prospectively validated this model using ECG images from a large, academic general hospital.

## 2. Materials and Methods

### 2.1. Study Population

A total of 2849 electrocardiograms constituted the study population, encompassing individuals allocated to the training, validation, and testing sets. All ECG images from GE 5500 (General Electric Company, Boston, MA, USA).

A total of 2849 ECG images were studied for the model development in all cohorts between 2009 and 2021. HCM (171 ECG images) or DCM (364 ECG images) patients aged 18 years or older of the Chinese PLA General Hospital who underwent both ECG and cardiovascular magnetic resonance (CMR) within 30 days were included in the study. CMR Ingenia CX (Philips Healthcare, Best, Netherlands). The clinical diagnosis of HCM was based on a CMR demonstration of left ventricular hypertrophy (LVH) with a maximum wall thickness of ≥15 mm anywhere in the left ventricle wall according to the 2023 European Society of Cardiology (ESC) guidelines [18,19,20]. Other causes of hypertrophy in adults with cardiac or systemic disease were excluded by CMR, such as hypertension, aortic stenosis, or storage disease [21]. DCM patients were defined within the 2023 ESC Guidelines [20]. Patients with ischemic DCM were excluded, as were other non-ischemic heart conditions, including severe valvular heart disease, infiltrative cardiomyopathy, and congenital heart disease. A total of 1582 age- and sex-matched healthy controls were selected from individuals with no cardiovascular disease and normal ECG readings at our Health Check Center during the same time period.

Temporal validation datasets between 1 January 2022 and 30 December 2023, which included an additional 170 HCM and 174 DCM patients from the same center, were prospectively included in the temporal validation. All subjects were validated previously by a detailed manual review of records by a cardiologist. A flowchart showing the datasets used for model training, valuation, testing, and temporal validation was created (Figure 1).

### 2.2. ECG Image Preprocessing and Validation

Each ECG image in the HCM/DCM was reviewed by three cardiologists blinded to HCM/DCM vs healthy controls group in order to document the following ECG features using pre-defined criteria: normal vs. abnormal ECG, atrial fibrillation or flutters, Abnormal Q, an ST-T change, and a right bundle branch block (RBBB). ECG images in all were standard 12-lead 10 s ECGs acquired in the supine position at a sampling rate of 500 Hz using a GE-Marquette ECG machine. Standardized high-resolution electrocardiographic imaging protocols with clinically validated data integrity ensured the acquisition of diagnostically reliable waveforms. Quality assurance metrics confirmed complete image preservation. Patients diagnosed with at least one ECG within this period were randomly assigned to the training, validation, and testing sets. In the training cohort, all electrocardiograms from HCM/DCM within the 30 day window before CMR were, for each patient, included to maximize the use of the data. For the independence of the validation, we chose only one ECG per patient to be included during the performance evaluation.

### 2.3. Model Method

We constructed a convolutional neural network model based on the Deep Residual (ResNet) [21]. For deep learning, the mode of feature extraction is a convolution operation expressed asy = Conv(x)(1)
where Conv(∙) is the convolution operation, representing the weighted sum of the input x, and y denotes the output vector.

In addition to the standard convolutional layer, ResNet includes a residual connection that directly links the input x to the output, followed by an element-wise addition of the convolution output Conv(x) and the input x to produce the final output. This residual connection was primarily designed to address the “degradation problem” encountered as deep convolutional networks increase in depth, namely, the issue of vanishing/exploding gradients, which is represented asH(x) = Conv(x) + x(2)

This residual structure effectively creates an identity mapping, guiding the network towards convergence along the identity path. This ensures that the overall error rate does not deteriorate with an increasing network depth and facilitates the extraction of latent image features that may be challenging for recognition.

The model used in our study is ResNet-50, which is based on ResNet architecture and consists of 50 layers. The architecture includes an initial convolutional layer followed by a series of residual blocks, each containing multiple convolutional layers with skip connections. These skip connections bypass one or more layers, allowing the network to learn identity mappings, which helps in training very deep networks. After the residual blocks, the network ends with a global average pooling layer, followed by a fully connected layer and a softmax output for classification tasks.

### 2.4. Model Training

We divided the model input images into three parts: the training set, the validation set, and the testing set, with the proportions of 60%, 10%, and 30%, respectively. Then, we preserved the natural aspect ratio of the ECG image and resized the original image to 400 × 256 pixels to expedite model processing. To enhance the model’s generalization ability and ensure the input image reflects real-world variability, each ECG image was randomly rotated by up to 30 degrees, and a 20-pixel square cutout was applied as a masking technique [22] (Figure 2).

The formal training of the model comprises two stages (Figure 3): pre-training and fine-tuning. Initially, the pre-trained model was trained using non-background ECG images for 200 epochs with an Adam optimizer and a learning rate of 0.001 [23]. By utilizing non-background ECG images, the model effectively focuses on the ECG waveform region while minimizing noise interference from the background grid. Subsequently, as depicted in Figure 3, the trained ResNet50 model was adopted, and all layers except the final linear layer, convolution layer, and residual layer were frozen to commence fine-tuning on the original ECG images. During this stage, the Adam optimizer was still employed with a reduced learning rate of 0.00001.

Additionally, both the pre-trained model and the formal model utilized a batch size of 128. During the training phase, if the validation loss did not decrease for three consecutive epochs, we implemented a 5% reduction in the learning rate following a linear rule. Given the limited size of HCM or DCM samples, clinical labels were amalgamated from multiple reports and subjectively influenced. To mitigate the risk of the model becoming overly confident in its predictions during training and to enhance its generalization capability, we employed label smoothing technology [24], which is expressed as follows:(3)$$y=\left\{\begin{array}{c} 1-\alpha ,\;if\;i=target\\ \frac{\alpha }{K},\;if\;i\;\neq \;target \end{array}\right.$$
where α is the sensitivity coefficient, and K is the number of categories. In this analysis, three experts in the field of cardiology were engaged to horizontally segment patients based on cardiac MRI findings and clinical signs and symptoms. Our model was validated using both an internal test set and temporal validation patients.

It is worth noting that the non-background pre-trained model only participates in the training phase. In the model diagnosis stage, only normal ECG images are needed to diagnose HCM/DCM.

After model training, the Grad-CAM (Gradient-weighted Class Activation Mapping) technique was used to visualize which regions of an ECG image are most important for the classification of DCM/HCM. It generates a heatmap, highlighting the areas that contribute most to the predicted class.

## 3. Results

This section describes the results of detection in HCM or DCM, and it also evaluates which method demonstrated superior performance. The median age of the subjects in the model development cohort was 41 years (interquartile range 30–51 years) at the time of the ECG recording. Among these, 1570 (55.1%) were male, and 1279 (44.9%) were female. Furthermore, with the Grad-CAM technique, the important regions were associated with the left ventricular.

### 3.1. Detection of DCM

In the test set comprising original standard ECG images, the AUROC for DCM was 0.996, and the AUPRC was 0.944 (Table 1, Figure 4). The model achieved a precision of 0.995 and a recall of 0.991. Furthermore, when focusing on ECG diagnoses with ST-T changes, the accuracy and recall reached 0.951 and 0.921, respectively. For abnormalities such as abnormal Q waves, the model achieved an accuracy and recall rate of 0.974 and 0.857, respectively. Similarly, for the high left ventricular voltage, the accuracy and recall were 0.923 and 0.812, respectively. Overall, the model effectively distinguished between DCM and healthy controls, with 11% of DCM patients predicted as HCM patients.

AUPRC indicates area under precision recall curve; and AUROC, area under receiver operating characteristic curve; RBBB, right bundle branch block.

### 3.2. Detection of HCM

In the standard ECG test set, the AUROC and AUPRC for the model’s detection of HCM were 0.980 and 0.951. The precision and recall were 0.925 and 0.877 (Table 1, Figure 4), respectively. Based on these results, the precision and recall for ECG diagnoses of ST-T changes, the abnormal Q-wave, and the high left ventricular voltage were comparable to those for HCM. For the right bundle branch block (RBBB), the precision and recall values were 1.0 and 0.833, respectively, which were slightly lower than those for HCM. Remarkably, the model achieved its highest accuracy for both dilated cardiomyopathy and hypertrophic cardiomyopathy in cases of atrial fibrillation, RBBB, and individuals aged over 65 years. Of the 859 tested electrocardiograms, 688 were normal electrocardiograms, accounting for 80.1%. The model AUPRC for samples in the normal electrocardiogram range reached 0.958 (DCM positive) and 0.907 (HCM positive), respectively (Figure 5). Overall, the model effectively distinguished between predictions of HCM and healthy controls, with 13% of HCM patients predicted as DCM patients (Figure 6). In addition, the model performed similarly regardless of age, sex, and ECG diagnosis (Table 1). 

### 3.3. Localization of Predictive Clues for HCM/DCM

Figure 7 shows activation heat maps for some of the HCM or DCM ECG image categories, showing the areas in the ECG image layout where the model is most confident in predicting cardiomyopathy, with red indicating high importance and blue indicating low importance. For standard 12-lead ECGs, the V2 lead is most distinctive for predicting DCM based on the presence of abnormal QRS waves (Figure 7A). Meanwhile, in combination with the V3 lead, the presence of high voltage is the most distinctive feature for predicting HCM (Figure 7B). Overall, the signal strength of Grad-CAM was higher in the V2 and V3 regions than in other regions of the ECG.

### 3.4. Temporal Validation

The model exhibited stability and reliability during temporal validation in patients. The performance of the model was consistent with the temporal validation datasets in HCM/DCM (Figure 8), which comprised 879 ECG images from consecutive patients, demonstrating an AUROC of 0.979/0.979 and an AUPRC of 0.710/0.878.

## 4. Discussion

In this study, we developed a fine-tuned ResNet50 for the detection of image-based HCM or DCM in a clinical 12-lead ECG. The ECG image-based model is highly discriminatory in detecting HCM cases, with an AUROC of 0.980. It distinguishes the leads across standard 12-lead ECG image formats, making it suitable for implementation in clinical care. The model’s higher discriminatory performance was in our elderly age cohort (≥65 years) in this study (AUROC = 1.0), in which the study excluded young adults (<18 years). Additionally, in the subgroup of ECG reports with the right bundle branch block (RBBB), the image-based model performed nearly identically to the overall population (AUROC = 1.0), indicating that the image model is useful for distinguishing HCM.

Previous research has concentrated on developing automated algorithms for detecting cardiomyopathy and characterizing its phenotype based on original ECG signal-based data; these studies also typically involve relatively small patient samples. These algorithms have utilized specific ECG features, for example, an elevated QRS wave voltage and abnormal Q waves or abnormal T waves, achieving performance levels comparable to interpretations made by experienced electrophysiologists.

ECG recordings are commonly stored in image formats and extensively used in clinical practice, making image-based tools particularly valuable in low-resource settings. Additionally, signal-based ECG data are often not preserved beyond the basic ECG images [25]. Despite advancements in multimodal cardiac imaging technologies, an ECG remains a crucial preliminary assessment examination for patients with HCM or DCM. Utilizing ECG images for HCM or DCM detecting represents a novel application of artificial intelligence (AI) with the potential to improve clinical care. In the early stages of these conditions, ECG abnormalities might be the only signs present [26]. Traditionally, ECG is valued for its high sensitivity and simplicity, offering insights into various disease conditions, especially in arrhythmias, left ventricular hypertrophy (LVH), or myocardial ischemia.

Hypertrophic cardiomyopathy (HCM) involves myocardial thickening and changes that are not solely due to an abnormal heart load. Its etiology is multifactorial, and the pathogenesis is complex. ECG manifestations of HCM include the following: 1. About 12–25% of patients may exhibit a completely normal ECG report, over 80% show non-specific ST-T changes, and a subset of those with apical hypertrophic cardiomyopathy may present with pronounced T-wave inversion [27,28]; 2. Left ventricular hypertrophy (LVH) and left bundle branch block (LBBB) are also common; 3. Abnormalities in Q waves. The basal segment of the anterior septum and the free wall of the anterior wall are the areas most commonly affected by LVH. In some populations, LVH may be only localized in one or two segments of the left ventricle, and in certain cases, right ventricular wall hypertrophy may also be observed [29]. Hypertrophic cardiomyopathy (HCM) can lead to severe symptoms or sudden cardiac death, particularly among young athletes. Previous research has concentrated on high-risk stratification based on HCM characteristics, specific ECG image features, and morphological ECG findings [30,31]. Recently, the Mayo Clinic developed an AI-based signal ECG model demonstrating high sensitivity and high accuracy in HCM. This deep learning AI model, utilizing 12-lead signal ECG data, has also shown high accuracy within a pediatric population (<18 years) [32]. Despite the presence of numerous abnormal ECG findings in HCM patients, determining which ECG variables are the most efficient and accurate for identifying HCM remains challenging. To address this, adaptive least absolute shrinkage and selection operator (LASSO) analysis identified only two key ECG features—T wave inversion and the amplitude of the S wave in lead V1—for inclusion in the predictive model. This model demonstrated an effective discriminative performance between HCM and non-HCM patients [33].

ECG examination is recommended as a class I diagnostic tool for all patients suspected of HCM or DCM, with an annual follow-up advised [34]. Despite various proposed ECG criteria for diagnosis, none have consistently demonstrated a reliable performance. For approximately 10% of HCM patients, an ECG may show a normal report, which can undermine the effectiveness of traditional diagnostic ECGs [35,36]. The AI-enabled ECG, however, offers a powerful tool for detecting HCM across diverse patient groups. The Mayo Clinic’s findings revealed that this model achieved robust accuracy in subjects with left ventricular hypertrophy (LVH) (AUC = 0.95) [37].

Dilated cardiomyopathy (DCM) is marked by impaired dilation and contraction of the left ventricle. ECG abnormalities are present in up to 80% of the patients with DCM [38]. The causes of DCM can be genetic etiology (primary DCM) or acquired etiology (secondary DCM). Acquired etiology includes infections, toxins, viral infections, alcohol use, cancer treatments, endocrinopathies, pregnancy, tachyarrhythmias, and immune-mediated diseases. About 5–15% of the patients with acquired DCM may have a likely relevant gene variant. Distinct ECG abnormalities may be linked to specific genetic or secondary forms of DCM [3]. ECG features associated with DCM include 1. High sensitivity, where a completely normal ECG can rule out DCM; 2. Complexity, with possible abnormalities in depolarization, repolarization, and heart rhythm; 3. Variability, where arrhythmias can change over time, particularly in the atrioventricular, and include bundle branch blockages. The ECG characteristics indicative of DCM are high voltage in the left chest lead, low voltage in the limb leads, and poor R-wave progression in the chest leads. These features are highly valuable for the diagnosis of DCM [39]. Notably, an elevated voltage in the left thoracic lead is particularly significant. Additionally, according to the Momiyama standard, an increased R_V6_/R_max_ value (>3) is a distinctive ECG marker that holds considerable diagnostic and prognostic value for DCM [40]. Moreover, ECG features such as lateral inverted T-waves, intraventricular conduction delay, low voltage, and fragmented QRS complexes have been independently associated with late gadolinium enhancement cardiac MRI (LGE-CMR) [41]. ROC analysis showed a notable accuracy in the model (AUC 0.66), where these ECG predictors were incorporated alongside the clinical-LVEF model. LGE-CMR is a reliable and reproducible method for evaluating myocardial fibrosis and is considered the gold standard for diagnosing cardiomyopathy [42,43,44,45]. While ECG has traditionally been viewed as non-specific for DCM, emerging insights into genotype–phenotype correlations offer the potential for identifying specific patterns and abnormalities that may highlight distinctions in DCM subtypes [46].

A previous AI-based ECG model has also shown high accuracy from multiple groups, but no study has demonstrated a consistent performance [47,48]. In this study, we explored the current research on 12-lead ECG images for diagnosing and managing DCM or HCM. HCM is characterized by T-wave inversions (TWIs) in ≥2 leads in Figure 7B, and is attributed to myocardial disarray and delayed repolarization secondary to asymmetric septal hypertrophy. HCM manifests asymmetric septal hypertrophy and a high voltage in the left ventricular (R-wave amplitude >2.6 mV in precordial leads) and reflects compensatory electrical remodeling in the thickened myocardium. In particular, strain-pattern ST depression with TWIs (>0.2 mV in lateral leads) correlates with microvascular ischemia. Notably, deep TWIs may exhibit strong discriminatory power, serving as electrophysiological hallmarks of myofiber disarray. A DCM prolonged QRS duration arises from diffuse myocardial-fibrosis-induced conduction delays, particularly in the left bundle branch system. Low-voltage QRS complexes (<0.5 mV limb leads) correlate with chamber dilatation and a reduced myocardial mass. Non-specific ST-T changes in Figure 7A, reflect interstitial fibrosis and mechanoelectrical dyssynchrony, which is distinct from the ischemic-like “strain” patterns in HCM.

We focused on the practical application of ECG images in clinical settings to identify disease presence, highlighting how ECG AI-based models can offer crucial insights and enhance the role of this fundamental technique as cardiomyopathies evolve. Our goal is to develop a simple, user-friendly, and highly functional pre-diagnosis model using common ECG images as a detecting tool for HCM or DCM. We suggest that the adoption of AI-enabled ECG could be beneficial, particularly in improving diagnostic accuracy for cases that are challenging to classify using standard clinical and imaging criteria.

In terms of interpretability, Grad-CAM analysis images show that for all 12-lead ECG images, the regions corresponding to the V2 and V3 leads (anterior or anteroseptal) are the most important regions for predicting HCM or DCM. This aligns with the anatomical regions typically associated with the clinical diagnosis of HCM or DCM. Detecting and diagnoses for HCM or DCM may be limited by high false-positive rates due to non-specific features or, more rarely, by ECGs that may be apparently normal. This was a single-center study with an HCM or DCM population that comprised a large proportion of referral patients; thus, applicability in other clinic area settings remains to be determined.

## 5. Limitations

This study has several limitations. Firstly, the model was used to establish a robust framework for detecting two major forms of cardiomyopathy (DCM/HCM) before expanding to rarer subtypes, limiting generalizability to populations with CAD or secondary LV hypertrophy. Secondly, validation in real-world cohorts may overlap with comorbidities. Future studies are needed to validate our results in diverse cohorts. Generalization of our findings to Restrictive Cardiomyopathy (RCM) or Arrhythmogenic Right Ventricular Cardiomyopathy (ARVC) requires further validation. We advise caution in extrapolating our conclusions to patients with complex comorbidities until additional multi-center evidence is available.

## 6. Conclusions

Based on the availability of real 12-lead ECG images in clinics, this study developed and temporally validated an image model that detects HCM or DCM. This image model demonstrates robust performance across different time periods in patients and provides an automated and accessible detecting strategy for patients with HCM or DCM, which may inform the early timing of imaging/interventions and facilitate improved access to care.

The model can be effectively applied to real clinic care, reducing the cost of cardiomyopathy diagnosis for patients who are not diagnosed. In addition, ECG model interpretation related to HCM/DCM 12-lead ECG images can provide clinicians with an auxiliary interpretation to improve the application in clinical practice.

## Figures and Tables

**Figure 1 bioengineering-12-00250-f001:**
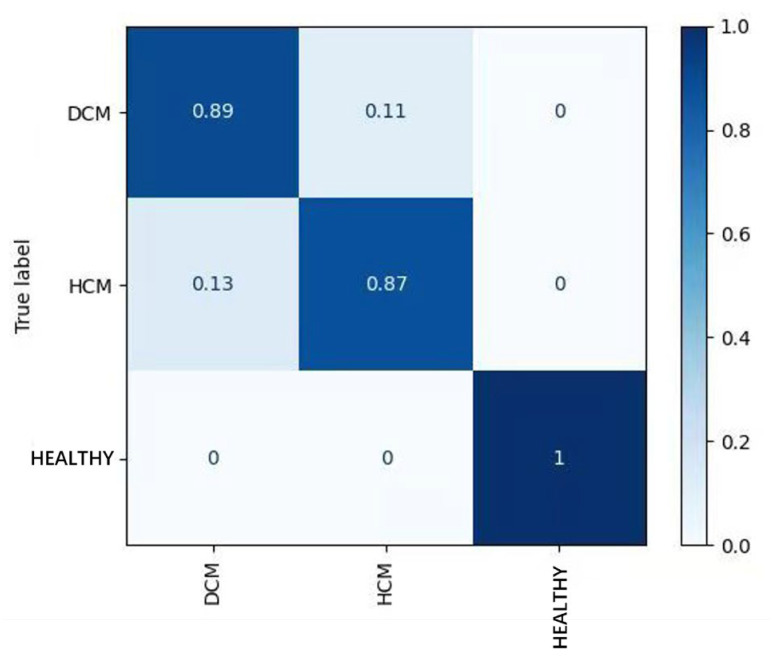
Data flowchart and model design of the study patients.

**Figure 2 bioengineering-12-00250-f002:**
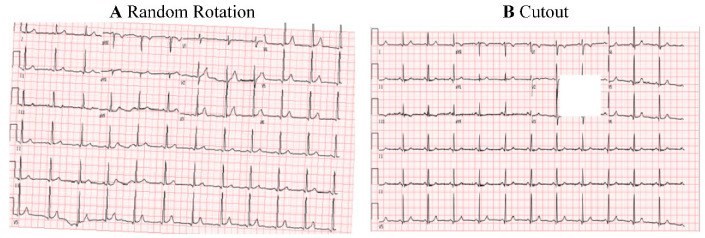
Electrocardiogram in two formats: random rotation and cutout. (**A**) The ECG image was randomly rotated by up to 30 degrees; (**B**) A 20-pixel square cutout from the ECG image was applied as a masking technique.

**Figure 3 bioengineering-12-00250-f003:**
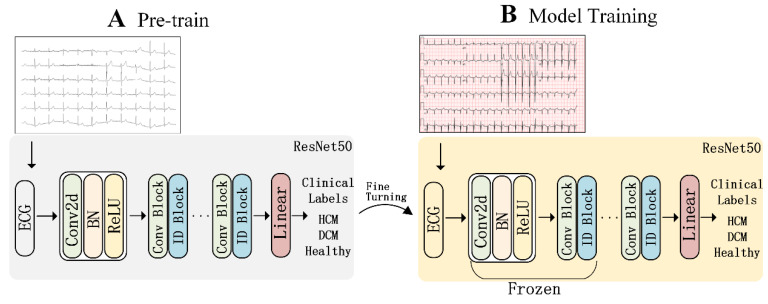
The main structure of the model includes pre-trained non-background ECG model and fine-tuned standard ECG. (**A**) The pre-trained model was trained using non-background ECG images for 200 epochs; (**B**) All layers from the trained ResNet50 model except the final linear layer, convolution layer, and residual layer were frozen to commence fine-tuning on the original ECG images.

**Figure 4 bioengineering-12-00250-f004:**
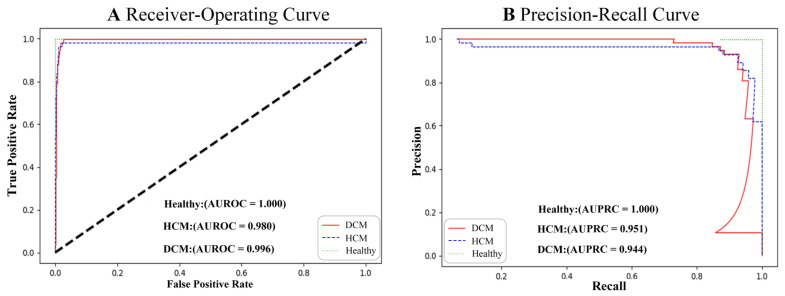
Receiver–operating (**A**) and precision–recall (**B**) curves on ECG images test datasets.

**Figure 5 bioengineering-12-00250-f005:**
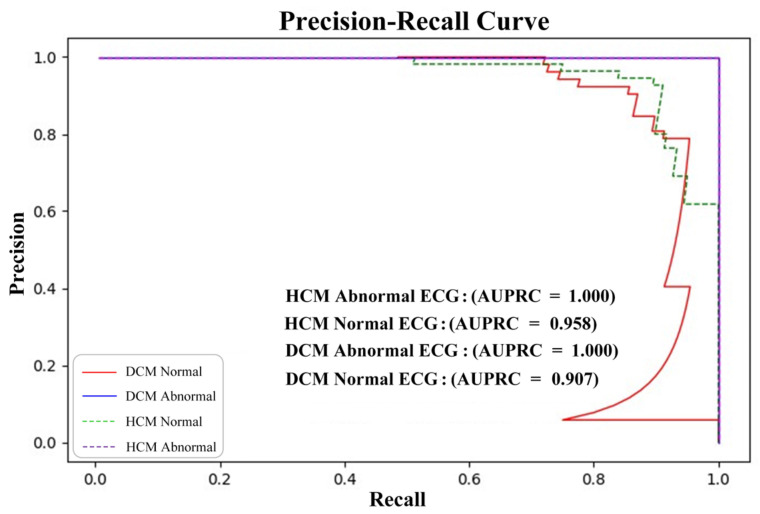
Precision–Recall curves on normal/abnormal ECG images.

**Figure 6 bioengineering-12-00250-f006:**
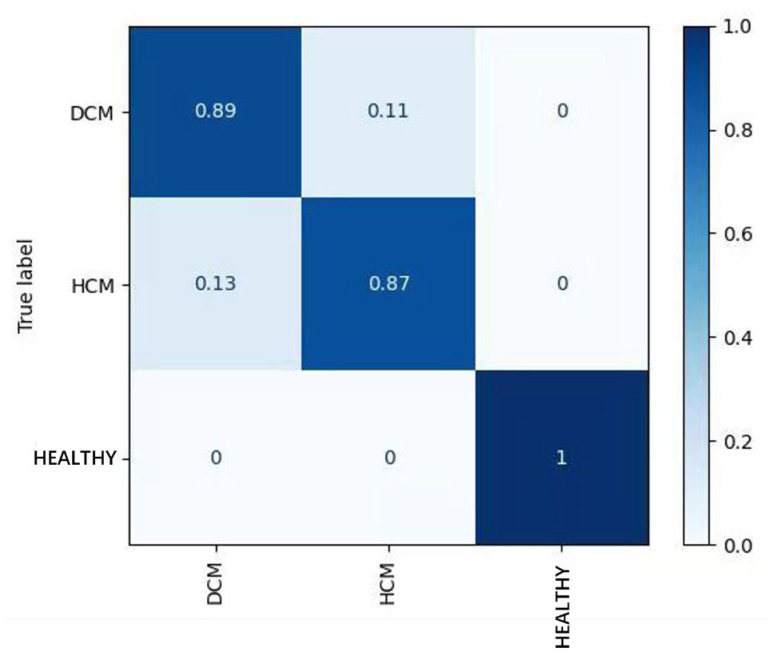
Confusion matrix of classification result.

**Figure 7 bioengineering-12-00250-f007:**
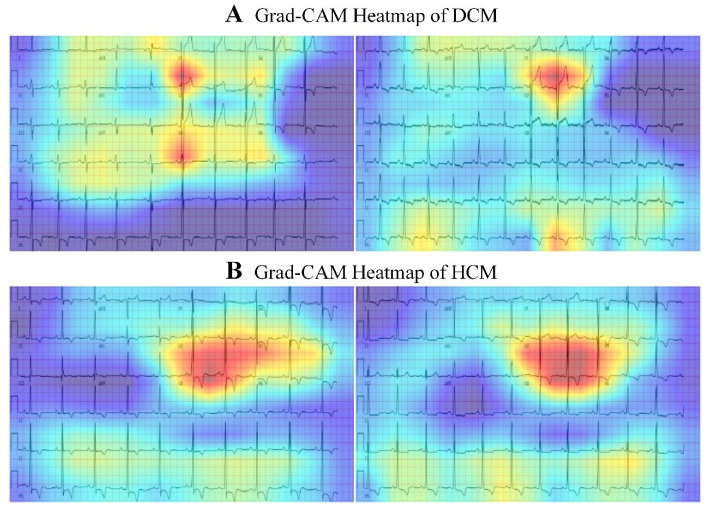
The heatmaps represent averages of the 100 positive cases with the most confident model predictions for DCM and HCM. (**A**) The V2- V3 lead is most distinctive for predicting DCM based on the presence of abnormal QRS waves; (**B**) In combination with the V2- V3 lead, the presence of high voltage is the most distinctive feature for predicting HCM.

**Figure 8 bioengineering-12-00250-f008:**
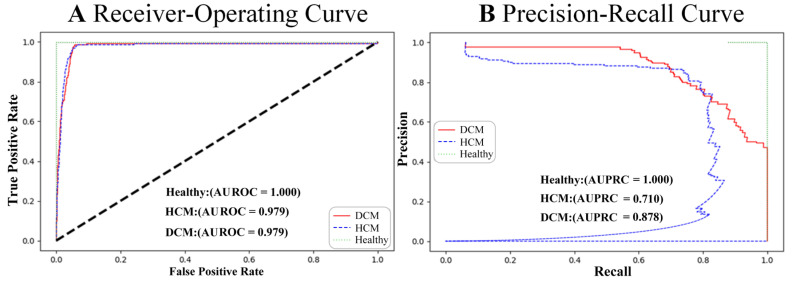
Receiver–operating (**A**) and precision–recall (**B**) curves on ECG images in validation patients.

**Table 1 bioengineering-12-00250-t001:** Performance of Model on Test Images Across Demographic Subgroups in the Test Set.

Labels (DCM/HCM)	No	Precision	Recall	F1	AUROC	AUPRC
All	859	0.995/0.925	0.991/0.877	0.991/0.900	0.996/0.980	0.944/0.951
Male	485	0.993/0.933	0.991/0.913	0.992/0.923	0.993/0.975	0.897/0.957
Female	374	0.997/0.9	0.994/0.818	0.995/0.857	0.998/0.998	0.955/0.943
≥65 y	56	1.0/1.0	0.959/0.777	0.979/0.875	1.0/1.0	1.0/1.0
<65 y	803	0.993/0.888	0.989/0.833	0.991/0.860	0.995/0.979	0.887/0.939
Atrial fibrillation or flutter	9	1.0/1.0	0.333/0.333	0.5/0.5	0.944/0.944	0.943/0.902
No atrial fibrillation or flutter	850	0.993/0.901	0.99/0.851	0.991/0.876	0.996/0.979	0.906/0.949
left ventricular high voltage	36	0.892/0.892	0.961/0.961	0.925/0.925	0.923/0.942	0.812/0.968
No left ventricular high voltage	823	0.996/0.892	0.992/0.806	0.994/0.847	0.997/0.937	0.911/0.918
Abnormal Q	13	0.857/0.857	0.857/0.857	0.923/0.920	0.974/0.980	0.857/0.857
No Abnormal Q	846	0.992/0.872	0.988/0.82	0.990/0.845	0.995/0.977	0.882/0.938
RBBB	16	1.0/1.0	0.888/0.833	0.941/0.909	1.0/1.0	1.0/1.0
No RBBB	843	0.991/0.865	0.992/0.883	0.991/0.883	0.996/0.978	0.889/0.937
ST-T change	59	0.941/0.941	0.888/0.888	0.914/0.914	0.951/0.956	0.921/0.972
No ST-T change	800	0.994/0.863	0.997/0.904	0.996/0.883	0.997/0.951	0.863/0.904

## Data Availability

The original data supporting the result will be made available by the authors.

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
