# Peer review of "Identifying Hypertrophic or Dilated Cardiomyopathy: Development and Validation of a Fine-Tuned ResNet50 Model Based on Electrocardiogram Image"

_bioengineering, 2025, doi:10.3390/bioengineering12030250_

Round 1
Reviewer 1 Report
Comments and Suggestions for Authors
ECG is a common primary investigation in cardiac patients. Being popular in the last century for investigations in organic heart diseases, in was replaced by echo and cmr later. AI technologies allow deeper look at ECG, both 12-leads and single-lead. Current paper presents results of ResNet50 -based algorithm for identifying HCM or DCM based on 12-lead ECG. The study is well -done, however the paper requires several clarifications.
1. The title presents the topic of the study. However, ResNet50 could be included it the title to clarify the research.
2. Abstract represented the text body.
3. Introduction section represents the scope of the problem. It is brief and can include more data on AI ECG-reading models. Moreover, the aim of the study could be more précised.
4. Materials and methods. Several issues should be clarified.
4.1 Study population. Flowchart is presented. Were patients with ischemic DCM excluded? Who were the controls? Were only patients with normal ECG included as controls?
4.2 Experiment design. The title has to be clarified. As so as the phrase “Each included ECG was clinically confirmed by three cardiologist within”. What was confirmed?
Model method and model training are clear.
5 Result section is good.
6.Discussion section.
Notable, Limitation section has to be incorporated. It the study only three cohorts: DCM, HCM and controls. That results could not be translated into daily practice with high prevalence of CAD with previous MI, LVH etc.
7.Conclusion sections should be included
8. Figures and tables are ok.
9. References are updated.
Author Response
We sincerely appreciate your valuable feedback and have carefully considered your concerns regarding this manuscript. Thank you for highlighting these critical points. We believe these revisions will strengthen the manuscript’s transparency and relevance to clinical practice. Below is our point-by-point response. Comments 1: The title presents the topic of the study. However, ResNet50 could be included in the title to clarify the research. Response 1: Thank you for this suggestion. We agree that specifying the methodology enhances clarity. We have revised the title to include "ResNet50". This change is reflected in the revised manuscript (Title, Line 1). Revised Title: " Identifying hypertrophic or dilated cardiomyopathy: Development and Validation of a fine-tuned ResNet50 Model Based on Electrocardiogram images" Comments 2: Abstract represented the text body. Response 2: We thank the reviewer for this observation. The abstract has been restructured to align precisely with the manuscript body. “There is no established detecting tool for hypertrophic cardiomyopathy (HCM) and dilated cardiomyopathy (DCM). This study aimed to develop a deep learning-based model for identifying HCM and DCM using standard 12-lead electrocardiogram (ECG) images. We obtained a cohort of patients with HCM (171 ECG images) or DCM (364 ECG images) confirmed by cardiovascular magnetic resonance (CMR) examinations, underwent both ECG and CMR within 30 days at our institution. Age- and sex-matched healthy controls (2314 ECG images) were selected from our Health Check Center. A total of 2,849 ECG images were processed via a fine-tuned ResNet50 architecture, with stratified 5-fold cross-validation for model training, validation and testing. The proposed model demonstrated strong performance in distinguishing DCM, achieving an area under the receiver operating curve (AUROC) of 0.996 and an area under the precision-recall curve (AUPRC) of 0.940. For the detection of HCM, the model also achieved an AUROC of 0.980 and an AUPRC of 0.953, respectively. The model prospectively exhibited stability in temporal validation. Furthermore, representative images of the Gradient-weighted Class Activation Mapping (Grad-CAM) technique analysis showed the region corresponding to the anterior and anteroseptal leads were the most important areas for prediction of HCM or DCM. This temporally validated fine-tuned ResNet50 model shows promise to inexpensively detect individuals with HCM or DCM.” Revised text is marked in red (Abstract, Lines 14-20). Comments 3: Introduction section represents the scope of the problem. It is brief and can include more data on AI ECG-reading models. Moreover, the aim of the study could be more precise. Response 3: We appreciate this feedback. The introduction now revised includes more data on existing AI-ECG models (Introduction, Lines 55-78; Page 2) and clarifies the study aim as Added text is highlighted in red: "To develop a ResNet50-driven framework for differentiating HCM or DCM from standard 12-lead 10-s ECG images, addressing gaps in screening tools for cardiomyopathies." Added text is highlighted in red (Introduction, Lines 81-82; Page 2). Comments4.1: Study population. Flowchart is presented. Were patients with ischemic DCM excluded? Who were the controls? Were only patients with normal ECG included as controls? Response 4.1: We thank the reviewer for highlighting this ambiguity. Flowchart also presented. The revised Methods section clarifies (2.1. Study Population, Lines 98-102):
Comments 4.2: Experiment design. The title has to be clarified. As well as the phrase “Each included ECG was clinically confirmed by three cardiologist within”. What was confirmed? Response 4.2: We apologize for the lack of clarity. The revised Methods section includes (Section 2.2, Lines 109-116): - Revised Subheading: " ECG Image Preprocessing and Validation: " Changes are highlighted in red. - Clarified Text: " Each ECG image in the HCM/DCM was reviewed by three cardiologists blinded to HCM/DCM vs. control in order to document the following ECG features using pre-defined criteria: normal vs. abnormal ECG, atrial fibrillation or flutter, Abnormal Q, ST-T change, right bundle branch block (RBBB). ECG images in all were standard, 12-lead 10-s ECGs acquired in the supine position at a sampling rate of 500 Hz using a GE-Marquette ECG machine. Standardized high-resolution electrocardiographic imaging protocols with clinically validated data integrity ensure acquisition of diagnostically reliable waveforms. Quality assurance metrics confirm complete image preservation." Comments 6: Discussion section. Notably, Limitation section has to be incorporated. The study only includes three cohorts: DCM, HCM, and controls. These results could not be translated into daily practice with high prevalence of CAD with previous MI, LVH etc. Response 6: We deeply appreciate this critical point. A dedicated "Limitations" subsection has been added to the Discussion, stating: " This study has several limitations. Firstly, this model was to establish a robust framework for detecting two major forms of cardiomyopathy (DCM/HCM) before expanding to rarer subtypes, limiting generalizability to populations with CAD or secondary LV hypertrophy. Secondly, validation in real-world cohorts may be with overlap-ping comorbidities. The need for future studies to validate our results in cohorts with diverse cohorts, and generalization to Restrictive cardiomyopathy (RCM) or Arrhythmogenic Right Ventricular Cardiomyopathy (ARVC) re-quires further validation. Caution in extrapolating conclusions to patients with complex comorbidities until further multi-center collaborations evidence is available." Marked in red (Limitations, Lines 368-376). Comments 7: Conclusion section should be included. Response 7: A standalone Conclusion section has been added Highlighted in red (Conclusion, Lines 378-387): "Based on the availability of real 12-lead ECG images in clinics, this study developed and temporally validated an image-model that detects HCM or DCM. This image-model is demonstrating robust performance across different time periods in patients, provides an automated and accessible screening strategy for patients with HCM or DCM, which may inform the early timing of imaging/interventions and facilitate improved access to care. The model can be effectively applied to real clinic care, reducing the cost of cardiomyopathy diagnosis for patients that are not diagnosed. In addition, ECG model interpretation related to HCM/DCM 12-lead ECG images can provide clinicians with auxiliary interpretation to improve the application in clinical practice.".
Response 8: Thank you for your valuable feedback. I have made the necessary adjustments to the figures and tables as suggested (Figure 4, Figure 5).
Response 9: References are updated Highlighted in red. |

Reviewer 2 Report
Comments and Suggestions for Authors
This paper develops and validates over time an image model tool for detecting HCM or DCM, and concludes that the image model demonstrates robust screening performance across different time periods for patients. As telemedicine advances, screening for heart disease using ECGs is a very important topic.
However, there are several problems with this paper, and it cannot be accepted as an Article category. You will need to either resubmit it under a different category, such as Report, or rewrite it substantially.
Hypertrophic cardiomyopathy (HCM) and dilated cardiomyopathy (DCM) are the two main types of cardiomyopathy, and as you point out, they are associated with a significant increase in morbidity and mortality.
The ECG waveform differs depending on the type of cardiomyopathy, but
DCM: LBBB, low-voltage QRS
HCM: left ventricular hypertrophy pattern, abnormal Q wave, giant negative T wave
I think this paper is concerned with the detection of these.
However, clinically,
RCM: low potential QRS, abnormal P wave, atrioventricular block
ARVC: ε wave, negative T wave in right chest leads
are also included in cardiomyopathy.
And for diagnosis, in addition to the ECG, additional tests such as echocardiography and MRI are also important.
The four main characteristics of dilated cardiomyopathy (DCM) are as follows.
Abnormal QRS wave: Left bundle branch block (LBBB) is common
ST-T changes: Non-specific ST changes, flattening or inversion of the T wave
Left ventricular hypertrophy pattern: Increase in the Sokolow-Lyon index (although the potential may be low for the degree of hypertrophy)
Abnormal Q wave: Can be seen even in the absence of ischemia
The four characteristics of hypertrophic cardiomyopathy (HCM) are as follows.
High-voltage QRS wave: Left ventricular hypertrophy pattern (deep S wave, high R wave)
Abnormal Q wave: Pseudo-infarction pattern (especially V5, V6, II, III, aVF)
Abnormal T wave: Giant negative T wave (especially in apical hypertrophic cardiomyopathy, V3-V5)
ST changes: ST depression (similar to ischemia)
Restrictive cardiomyopathy (RCM) is characterised by
low-voltage QRS: particularly low-voltage in limb leads
abnormal P-wave: left atrial load findings (wide, biphasic P-wave)
ST-T changes: negative T-wave
AV block: PQ prolongation or complete AV block
and,
the characteristics of Arrhythmogenic Right Ventricular Cardiomyopathy (ARVC) are
Epsilon wave (ε wave): a small wave immediately after the QRS in V1-V3 (suggesting abnormal conduction in the right ventricle)
T wave inversion: seen in the right chest leads (V1-V3)
QRS wave prolongation: often seen in right bundle branch block (RBBB) patterns
So there are several things in common.
However, your paper only concludes that ‘HCM or DCM is detected’, and there is no comparison with RCM or ARVC, and there is no detailed discussion on the differentiation between DCM and HCM.
◼︎ Minor points
Please check the format of the figures again. The text in the figures is small, so
please correct it to make it larger and easier to read.
Author Response
We sincerely appreciate your valuable feedback and have carefully considered your concerns regarding this manuscript. Thank you for highlighting these critical points. We believe these revisions will strengthen the manuscript’s transparency and relevance to clinical practice. Below is our point-by-point response. The figures was replace.
And for diagnosis, in addition to the ECG, additional tests such as echocardiography and MRI are also important.
Differentiation between DCM and HCM also show in RED text (4. Discussion, Lines 339-350) as below.
In this study, we explore the current research on 12-lead ECG image for diagnosing and managing DCM or HCM. HCM is characterized by T-wave inversions (TWIs) in ≥ 2 leads in Figure 7. B , attributed to myocardial disarray and delayed repolarization secondary to asymmetric septal hypertrophy. HCM manifests asymmetric septal hypertrophy, left ventricular high voltage (R-wave amplitude >2.6 mV in precordial leads) reflects compensatory electrical remodeling in thickened myocardium. Specially, strain-pattern ST depression with TWIs (>0.2 mV in lateral leads) correlates with microvascular ischemia. Notably, deep TWIs may be exhibit strong discriminatory power, serving as electro-physiological hallmarks of myofiber disarray. DCM prolonged QRS duration arises from diffuse myocardial fibrosis-induced conduction delays, particularly in the left bundle branch system. Low-voltage QRS complexes (<0.5 mV limb leads) correlate with chamber dilatation and reduced myocardial mass. Nonspecific ST-T changes Figure 7. A reflect interstitial fibrosis and mechanoelectrical dyssynchrony, distinct from the ischemic-like "strain" patterns in HCM.
Response 1:
We sincerely appreciate the reviewer’s acknowledgment of echocardiography’s pivotal role in cardiomyopathy screening. Our decision to exclude echocardiographic data aligns with the study’s specific objectives.
Rationale for Prioritizing CMR Over Echocardiography:
While echocardiography remains a cornerstone for initial HCM/DCM evaluation due to its accessibility, this study utilized cardiac magnetic resonance (CMR) as the gold standard for inclusion:
- Accuracy Constraints: Echocardiographic measurements are prone to variability from operator-dependent techniques, anatomical complexities, and suboptimal acoustic windows in obese patients, leading to underdiagnosis rates.
- Diagnostic Superiority of CMR: CMR provides myocardial fibrosis, precise wall thickness mapping, and differentiation of HCM mimics. This ensures standardized phenotyping, reducing misclassification risks inherent to echocardiography.
- Screening Feasibility: AI-ECG models are designed to replace echocardiography- or CMR-based population screening, particularly in low-resource settings where echocardiography expertise is scarce.
However, your paper only concludes that ‘HCM or DCM is detected’, and there is no comparison with RCM or ARVC, and there is no detailed discussion on the differentiation between DCM and HCM.
Response: We thank the reviewer for highlighting the scope of cardiomyopathy subtyping. Our focus on HCM or DCM reflects the following evidence-based considerations:
1.Prevalence and Clinical Urgency:
HCM has a global prevalence of 1:200~500 and is underdiagnosed in ≥80% of cases, necessitating scalable screening tools. DCM constitutes ~40% of non-ischemic cardiomyopathies with a 1-year mortality of 25%, demanding urgent diagnostic differentiation. RCM (prevalence <1:10,000) and ARVC (1:5,000) represent rare phenotypes, with insufficient confirmed cases (<20 in our cohort) for reliable model training.
2.ECG Phenotypic Overlap:
Low QRS voltage in RCM overlaps with late-stage DCM, while ARVC primarily involves right ventricular abnormalities. Integrating these subtypes introduces classification noise without multi-center validation to ensure robustness.
3.Clinical Decision-Making Impact:
HCM/DCM differentiation directly guides therapy (e.g., ICD implantation, family screening), whereas RCM/ARVC management hinges on genetic testing and invasive assessments. Prioritizing HCM/DCM aligns with optimizing care pathways for the largest affected populations.
Future work we will expand subtype inclusivity through collaborative cohort enrichment.
References
PEREZ-SERRA A, TORO R, SARQUELLA-BRUGADA G, et al. Genetic basis of dilated cardiomyopathy [J]. Int J Cardiol, 2016, 224(461-72.
AKINRINADE O, OLLILA L, VATTULAINEN S, et al. Genetics and genotype-phenotype correlations in Finnish patients with dilated cardiomyopathy [J]. Eur Heart J, 2015, 36(34): 2327-37.

Reviewer 3 Report
Comments and Suggestions for Authors
I find the manuscript interesting and would like to honestly express my congratulations and admiration to the team for their study.
However, there are certain aspects I would like to address:
- Why is the cardiac ultrasound not discussed, since it is a widely available technique with a great utility in CMH and CMD?
- I personally consider ECG trace to be a better term than ECG image.
- Please formulate a conclusion to underline the practical aspects of your research.
- In the references you cited the
2014 ESC Guidelines on diagnosis and management of hypertrophic cardiomyopathy, however the European Society of Cardiology has released a newer edition in 2023 of the guidelines, which would be better to be referenced.
https://www.escardio.org/Guidelines/Clinical-Practice-Guidelines/Cardiomyopathy-Guidelines
Author Response
We sincerely appreciate your valuable feedback and have carefully considered your concerns regarding this manuscript. Thank you for highlighting these critical points. We believe these revisions will strengthen the manuscript’s transparency and relevance to clinical practice. Below is our point-by-point response. Comments and Suggestions for Authors I find the manuscript interesting and would like to honestly express my congratulations and admiration to the team for their study. However, there are certain aspects I would like to address: Comments 1: Why is the cardiac ultrasound not discussed, since it is a widely available technique with a great utility in CMH and CMD? Response 1: We sincerely appreciate the reviewer’s acknowledgment of echocardiography pivotal role in cardiomyopathy screening. Our decision to exclude echocardiographic data aligns with the study’s specific objectives and practical considerations: 1.Rationale for Prioritizing CMR Over Echocardiography: While echocardiography remains a cornerstone for initial HCM/DCM evaluation due to its accessibility, this study utilized cardiac magnetic resonance (CMR) as the gold standard for inclusion. Accuracy Constraints: Echocardiographic measurements are prone to variability from operator-dependent techniques, anatomical complexities, and suboptimal acoustic windows in obese patients, leading to underdiagnosis rates. 2.Diagnostic Superiority of CMR: CMR provides myocardial fibrosis, precise wall thickness mapping, and differentiation of HCM mimics. This ensures standardized phenotyping, reducing misclassification risks inherent to echocardiography. 3.Detecting Feasibility: AI-ECG models are designed to replace echocardiography- or CMR-based population screening, particularly in low-resource settings where echocardiography expertise is scarce.
Comments 2: I personally consider ECG trace to be a better term than ECG image. Response 2: We appreciate this feedback. The Introduction now includes expanded discussions on existing AI-ECG trace models and image model. Added text is highlighted in red (Introduction, Lines 68-78; Page 2). However, the raw signals ECG data is difficult to be saved and does not conform to the actual clinical care. Based on clinician observation needs, some research attempt to use ECG images to detect heart disease. Recent advances in deep learning have enhanced etiological differentiation of cardiomyopathies. Haimovich et al. (2023) developed a 12-lead ECG-based CNN achieving AUC 0.92 for distinguishing hypertrophic cardiomyopathy (HCM) from other left ventricular hypertrophy etiologies. Tayal et al. (2022) implemented a random forest model integrating clinical, genetic, and cardiac MRI data to subclassify dilated cardiomyopathy (DCM) into fibrosis-metabolic and non-fibrotic subtypes. Shrivastava et al. (2021) demonstrated a CNN model with 98.8% sensitivity and 100% negative predictive value for DCM detection (LVEF≤45%), vali-dated across multiple cohorts (AUC 0.955-0.98). Zhou et al. (2023) employed XGBoost with clinical/echocardiographic data to differentiate ischemic vs. non-ischemic DCM (AUC 0.934). Comparative ML analyses revealed random forest models with mini-mum redundancy maximum relevance achieved 91.2% accuracy in distinguishing DCM, HCM, and healthy subjects (average AUC 0.98).
Comments 3: Please formulate a conclusion to underline the practical aspects of your research. Response 3: Based on the availability of real 12-lead ECG images in clinics, this study developed and temporally validated an image-model that detects HCM or DCM. This image-model is demonstrating robust performance across different time periods in patients, provides an automated and accessible screening strategy for patients with HCM or DCM, which may inform the early timing of imaging/interventions and facilitate improved access to care. The model may be effectively applied to real clinic care, reducing the cost of cardiomyopathy diagnosis for patients that are not diagnosed. In addition, ECG model interpretation related to HCM/DCM 12-lead ECG images can provide clinicians with auxiliary interpretation to improve the application in clinical practice.
In the references you cited the 2014 ESC Guidelines on diagnosis and management of hypertrophic cardiomyopathy, however the European Society of Cardiology has released a newer edition in 2023 of the guidelines, which would be better to be referenced. |
Response : We sincerely appreciate the reviewer’s acknowledgment.

Round 2
Reviewer 1 Report
Comments and Suggestions for Authors
The paper has been improved substenitally and can be recommended to the Journal
Reviewer 2 Report
Comments and Suggestions for Authors
This manuscript has been corrected, so it will end it here. Thank you for your hard work.
Comments on the Quality of English LanguageNone